

# A model for HIV disclosure of a parent's and/or a child's illness

Grace Gachanja[1] and Gary J. Burkholder[1,2,3]

[1] College of Health Sciences, Walden University, Minneapolis, MN, United States
[2] College of Social and Behavioral Sciences, Walden University, Minneapolis, MN, United States
[3] Laureate Education, Inc., Baltimore, MD, United States

## ABSTRACT

HIV prevalence in Kenya remains steady at 5.6% for adults 15 years and older, and 0.9% among children aged below 14 years. Parents and children are known to practice unprotected sex, which has implications for continued HIV spread within the country. Additionally, due to increased accessibility of antiretroviral therapy, more HIV-positive persons are living longer. Therefore, the need for HIV disclosure of a parent's and/or a child's HIV status within the country will continue for years to come. We conducted a qualitative phenomenological study to understand the entire process of disclosure from the time of initial HIV diagnosis of an index person within an HIV-affected family, to the time of full disclosure of a parent's and/or a child's HIV status to one or more HIV-positive, negative, or untested children within these households. Participants were purposively selected and included 16 HIV-positive parents, seven HIV-positive children, six healthcare professionals (physician, clinical officer, psychologist, registered nurse, social worker, and a peer educator), and five HIV-negative children. All participants underwent an in-depth individualized semistructured interview that was digitally recorded. Interviews were transcribed and analyzed in NVivo 8 using the modified Van Kaam method. Six themes emerged from the data indicating that factors such as HIV testing, living with HIV, evolution of disclosure, questions, emotions, benefits, and consequences of disclosure interact with each other and either impede or facilitate the HIV disclosure process. Kenya currently does not have guidelines for HIV disclosure of a parent's and/or a child's HIV status. HIV disclosure is a process that may result in poor outcomes in both parents and children. Therefore, understanding how these factors affect the disclosure process is key to achieving optimal disclosure outcomes in both parents and children. To this end, we propose an HIV disclosure model incorporating these six themes that is geared at helping healthcare professionals provide routine, clinic-based, targeted, disclosure-related counseling/advice and services to HIV-positive parents and their HIV-positive, HIV-negative, and untested children during the HIV disclosure process. The model should help improve HIV disclosure levels within HIV-affected households. Future researchers should test the utility and viability of our HIV disclosure model in different settings and cultures.

Corresponding author
Grace Gachanja,
g_gachanja@hotmail.com

## BACKGROUND

The HIV pandemic continues to heavily affect Sub-Saharan African (SSA) countries (*UNAIDS, 2014*). In 2012, Kenya had an estimated 1.2 million adults aged 15–64 years infected with HIV with a prevalence of 5.6% among this age group (*NASCOP, 2014*). The prevalence of HIV among children aged 18 months to 14 years was 0.9% for an estimated 101,000 children living with HIV. Five percent of couples were serodiscordant; only 24% of their infected partners were on antiretroviral therapy (ART), and just 56% of those on ART were virologically suppressed (*NASCOP, 2014*). Children in Kenya are known to initiate and practice risky sexual behavior at an early age (*NASCOP, 2014*); *Defo & Dimbuene (2012)* reported a median age of sexual debut within Africa to be 18 years. Adults have also been reported to have unsafe sexual behavioral practices accompanied by nondisclosure of an HIV status (*Luchters et al., 2008*). These facts raise implications for the continued spread of the disease among parents and children. This information also suggests the importance of disclosure of HIV status in order to minimize the spread of the disease.

HIV disclosure has many benefits for parents (*Delaney, Serovich & Lim, 2008*; *Qiao, Li & Stanton, 2013*; *Rwemisisi et al., 2008*; *Vreeman et al., 2013*; *Wiener et al., 2007*) and children (*Lee & Rotheram-Borus, 2002*; *Qiao, Li & Stanton, 2013*; *Shafer et al., 2001*; *Vallerand et al., 2005*; *Vreeman et al., 2013*) such as stress relief, improved psychological health, trust, family bonding, and increased children's awareness of HIV. HIV diagnosis and linkage to care promotes HIV testing, diagnosis, and disclosure of other family members' HIV statuses (*Kulzer et al., 2012*). In Kenya, subsequent HIV testing of family members of index (first person diagnosed in a family) adults diagnosed with HIV, indicated that the infection rate was 71% and 18% among their partners and children respectively (*Kulzer et al., 2012*). High levels of stigma and discrimination affect HIV testing for adults and children (*Human Right Watch, 2008*; *Turan et al., 2011*) and reduce HIV disclosure of a parent's and a child's HIV status within the country (*Gachanja, Burkholder & Ferraro, 2014a*; *Gachanja, Burkholder & Ferraro, 2014b*; *Vreeman et al., 2015*).

HIV disclosure is widely known to be challenging for HIV-positive parents and healthcare professionals. Understanding and executing the disclosure process appropriately is key to assuring good outcomes in both parents and children. HIV disclosure is an emotional experience for parents (*Delaney, Serovich & Lim, 2008*; *Kennedy et al., 2010*; *Rochat et al., 2014*; *Qiao, Li & Stanton, 2013*; *Vreeman et al., 2013*) and their children (*John-Stewart et al., 2013*; *Petersen et al., 2010*; *Qiao, Li & Stanton, 2013*; *Vaz et al., 2011*; *Vaz et al., 2010*; *Vreeman et al., 2013*). HIV disclosure can result in negative effects in both parents (*Lee & Rotheram-Borus, 2002*; *Shafer et al., 2001*; *Qiao, Li & Stanton, 2013*; *Vreeman et al., 2013*; *Wiener, Battles & Heilman, 1998*) and children, especially teenagers (*Petersen et al., 2010*; *Oberdorfer et al., 2006*; *Qiao, Li & Stanton, 2013*; *Vaz et al., 2010*; *Vreeman et al., 2013*) such as depression, sadness, behavior problems, and increased stigma and stressful life events.

There are various models of HIV disclosure for a parent's and a child's HIV status. Key models for HIV disclosure of a parent's HIV status include the consequence theory

of HIV disclosure (*Serovich, 2001*) and the disclosure process model (*Chaudoir, Fisher & Simoni, 2011*; *Qiao, Li & Stanton, 2013*). The consequence theory of HIV disclosure stipulates that parents proceed with disclosure when they perceive the benefits to outweigh the consequences (*Serovich, 2001*). The disclosure process model (*Chaudoir, Fisher & Simoni, 2011*; *Qiao, Li & Stanton, 2013*) aims to maximize the positive outcomes of disclosure while limiting negative outcomes in both parents and children. Some key models for disclosure of a child's HIV status include the four-phase model (*Tasker, 1992*), the *Blasini et al. (2004)* model, and a recently published model by *Lowenthal et al. (2014)*. The four-phase model (*Tasker, 1992*) stipulates that parents go through four disclosure phases— secrecy, exploratory, readiness, and full disclosure. The *Blasini et al. (2004)* disclosure model involves five steps—disclosure training for healthcare professionals, parental preparation, individualized interactive assessment sessions, the disclosure event, and post disclosure support. *Lowenthal et al. (2014)* described a pediatric HIV disclosure model in which HIV-positive children were taught about HIV in a progressive developmental process within a clinic setting until they were fully disclosed to.

The published literature on HIV disclosure has reported separately on disclosure of HIV-positive children's and HIV-positive parents' HIV statuses to HIV-positive and HIV-negative children, respectively. In addition, the HIV disclosure process following diagnosis of an index parent's or child's HIV status, to the time of full disclosure of this and other subsequently diagnosed family members' HIV statuses to all children within an HIV-affected family is not well reported in the literature. A typical HIV-affected family in Kenya has many HIV-positive family members who may include both parent(s) and child(ren) (*Gachanja, Burkholder & Ferraro, 2014a*; *Gachanja, Burkholder & Ferraro, 2014b*; *Republic of Kenya, 2009*). At this time, Kenya does not have guidelines for disclosure of a parent's and/or a child's HIV status. Prior researchers in the country have consistently called for guidelines and training programs for healthcare professionals working with HIV-affected families (*Gachanja, Burkholder & Ferraro, 2014a*; *Gachanja, Burkholder & Ferraro, 2014b*; *John-Stewart et al., 2013*; *Turissini et al., 2013*). To comprehend the start-to-end process of HIV disclosure and address the gap in knowledge about the process of disclosure of a parent's and/or a child's HIV status within HIV-affected Kenyan families, we conducted a study to understand the lived experiences of HIV-positive parents and their children going through the disclosure process. Based on the results of the study, we propose a new HIV disclosure model for use by healthcare professionals to guide HIV-positive parents through disclosure of both a parent's and a child's HIV status following diagnosis of an index person within an HIV-affected family.

## METHODS

### Participant recruitment

The study was conducted at the Kenyatta National Hospital Comprehensive Care Center located near the city center of Nairobi, Kenya between December 2010 and January 2011. Thirty-four participants consisting of 16 HIV-positive parents, seven HIV-positive children, six healthcare professionals, and five HIV-negative children were purposively

recruited into the study by the first author. HIV-positive parents were selected because they had biological children aged 8–17 years to whom they had performed no, partial, or full disclosure of the parent's and/or an HIV-positive child's HIV status within their households. HIV-positive and HIV-negative children were selected because they were between 8 and 17 years old, and they had already received partial or full disclosure of their own and their parents' HIV statuses, respectively. Healthcare professionals were selected from diverse professions that provide healthcare and support services to HIV-affected families during the disclosure process. They included a medical doctor, clinical officer, psychologist, registered nurse, social worker, and a peer educator.

Recruitment was initially difficult; 10 potential participants approached by the first author refused to participate because they were unfamiliar with her. Recruitment improved after involving the clinic's clinical officers, registered nurses, and peer educators, who referred and introduced HIV-positive parents meeting the criteria for study participation to the first author. HIV-positive parents and HIV-positive children were recruited first during the clinic's morning operating hours. HIV-positive parents meeting criteria to participate in the study were purposively recruited within the reception, triage, and waiting rooms of the clinic during their regularly scheduled clinic visits. With the consent of their parents, we approached and recruited HIV-positive children on the second floor waiting room area of the clinic designated for family-oriented healthcare. Healthcare professionals were approached for study participation in their working areas at the clinic during the slower afternoon clinic operating hours. Towards the end of the recruitment period, a different set of HIV-positive parents were approached and requested to bring their HIV-negative children who met criteria to be in the study to the clinic for study participation. Recruitment was continued until interview data saturation was achieved.

At the time of recruitment, HIV-positive parents, healthcare professionals, and HIV-positive and negative children (along with their parents) were offered an explanation of the study. Those who agreed to participate were taken to a private room in the clinic where informed consent procedures were performed. HIV-positive parents and healthcare professionals who agreed to participate provided written informed consent; HIV-positive and HIV-negative children who agreed to participate provided written assent, and their parents provided written informed consent. Ethics approval for the study was received from the university's Institutional Review Board (Approval # 11-10-10-03904) and from the Kenyatta National Hospital Research Standards and Ethics Committee (Approval # P373/10/2010).

## Data collection and analysis

Each participant was engaged in a digitally recorded, in-depth semistructured interview conducted by the first author. Interview guides were obtained from a study conducted in the Democratic Republic of Congo and adapted with the authors' permissions (*Vaz et al., 2010*). The interview questions for HIV-positive parents, HIV-positive children, and HIV-negative children explored HIV testing, the disclosure process and timing, and the benefits and consequences of HIV disclosure. Healthcare professionals were asked similar questions and about their opinions on how HIV disclosure of a parent's and child's HIV

status should be approached. All children assented to be interviewed alone; their parents were offered the opportunity to be in the room when the children were interviewed, but none chose to do so. At the beginning of their interviews, children with partial disclosure were asked what illness they or their parents were being treated for; the illness mentioned (e.g., tuberculosis) was substituted for HIV in subsequent interview questions. Participant interviews lasted anywhere from 30 to 90 min.

Recorded interviews were transcribed by the first author and a local Kenyan university student trained in transcription. Transcripts were sent to five participants (HIV-positive parents and healthcare professionals) to check for transcription accuracy. Transcribed interviews were transferred into NVivo 8 for analysis. Phenomenological data analysis was performed using the modified Van Kaam method by the first author (*Moustakas, 1994*). Analysis proceeded with preliminary listing and grouping of the transcripts and reduction and elimination of irrelevant information. Similar information within the transcripts was clustered and coded using codes consistent with prior HIV disclosure research (i.e., the process of disclosure, stigma, discrimination, benefits, consequences, and psychological effects of disclosure). The codes were listed, summarized, and grouped into six emergent themes spanning the entire HIV disclosure process. To ensure inter-coder accuracy, the codes and themes were crosschecked against the transcribed data by the last author.

## RESULTS

The six themes that emerged from the data are displayed in Fig. 1. They included motivation for HIV testing, living with HIV, evolution of disclosure, questions associated with disclosure, emotions associated with disclosure, and the benefits and consequences of disclosure. The participants' demographic profiles are presented in Table 1. The themes are further described below; to assure confidentiality for the participants, pseudonyms have been provided for each quote.

### Motivation for HIV testing

HIV-positive parents, HIV-positive children, and HIV-negative children hailed from families composed of a variety of infected, uninfected, and untested living and deceased family members as displayed in Table 2. With the exception of one family, parents were the index persons diagnosed within each family. After diagnosis of the index person in a family, and as part of the HIV disclosure process, healthcare professionals explained that they typically counseled HIV-positive parents to bring all other family members in for HIV testing. HIV-positive parents explained that after receiving this advice, they sought testing for their spouses immediately to three years later. A few parents stated that their spouses, although seriously sick, refused to seek testing, which resulted in death from lack of treatment. Testing for children was delayed until spouses were tested. Some parents stated that they tested all their children, others waited years and then tested only the children they suspected to be infected, and one had not tested any of her children at the time of study participation. Hussein, an HIV-positive parent stated:

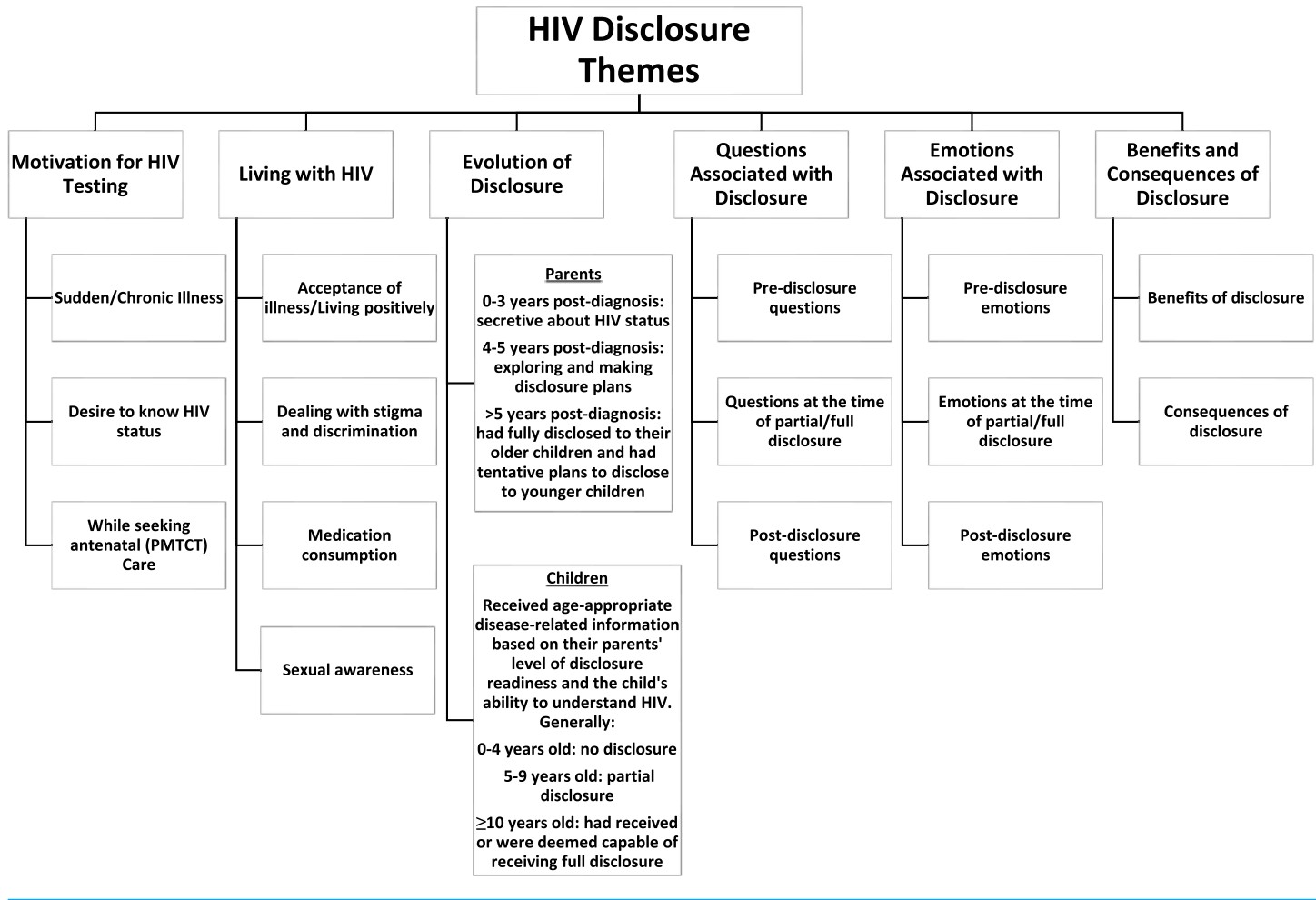

**Figure 1  HIV disclosure themes.**

[Clicks tongue] *My wife and I went to the doctor immediately when I became sick. She tested positive three years later when she became pregnant…* [Clicks tongue] *I never thought of testing the children because* [clicks tongue], *because of uh, okay I was stressed and then* [clicks tongue], *I did not have even time to consult them and they were a bit healthy, healthier than me. So I did this, I waited for time to come.* [Four years after his diagnosis] *they* [his two middle sons] *developed the symptoms, you know I knew the symptoms of this disease, I detected that there is something wrong, and okay there was that campaign of saying if somebody has TB, you have to go for the test, so we had to go for testing. I have five children, we started testing all of them.*

The most cited reason for initial testing of the index person within the represented families was a chronic or sudden illness presenting with AIDS-related symptoms. Four HIV-negative children explained that their parents had been diagnosed following chronic illnesses; these children were subsequently tested and found to be HIV-negative. Most HIV-positive parents and HIV-positive children explained that they had been tested after

**Table 1  Participants' demographic profiles.**

| Variable | Frequency | | | |
|---|---|---|---|---|
| | HIV-positive children | HIV-negative children | HIV-positive parents | Healthcare professionals |
| **Age** | | | | |
| **Children** | | | | |
| 12–13 | 2 | 1 | | |
| 14–15 | 1 | 3 | | |
| 16–17 | 4 | 1 | | |
| **Adults** | | | | |
| 21–30 | | | | 1 |
| 31–40 | | | 8 | 2 |
| 41–50 | | | 7 | 3 |
| 51–60 | | | 1 | |
| **Gender** | | | | |
| Female | 3 | 3 | 11 | 4 |
| Male | 4 | 2 | 5 | 2 |
| **Educational status** | | | | |
| Primary | 2 | 3 | 2 | |
| Secondary | 5 | 2 | 7 | |
| College | | | 7 | 6 |
| **Marital status** | | | | |
| Single | | | 1 | 2 |
| Divorced | | | 1 | |
| Widowed | | | 4 | |
| Married | | | 10 | 4 |

being chronically ill for a long time. Some married mothers were tested while pregnant and undergoing antenatal care; one HIV-negative child was also tested after her mother was diagnosed HIV-positive while pregnant with her younger brother. A few HIV-positive parents sought testing because they had wanted to know their HIV statuses, especially some married women, who suspected that their spouses were being unfaithful. Mary, an HIV-positive parent stated the following:

> I thought it's good to know my status, so I needed to know. And the reason why I used to seek testing, I used to doubt my husband [laughs]. He was having an affair outside, I had a good reason to do it, so that's why I used to go. And when I asked him if we go together, he always resisted.

## Living with HIV

After an HIV-positive diagnosis, healthcare professionals explained that they provided counseling to the infected person to help with acceptance of an HIV status. HIV-positive parents and HIV-positive children reported that they accepted that they were HIV-positive anywhere from immediately to many years later; HIV-negative children expressed that they accepted that their parents were HIV-positive soon after they received partial or

**Table 2 HIV-positive children's, HIV-negative children's, and HIV-positive parents' family members testing and HIV statuses.**

| Participant | HIV-positive children | HIV-negative children | HIV-positive parents |
|---|---|---|---|
| Family A | Parent: single HIV-positive mother<br>Siblings: 5 HIV-negative | Parents: 1 deceased, 1 HIV-positive<br>Siblings: 3 untested, 1 HIV-negative | Spouse deceased and HIV-positive<br>Children: 1 HIV-positive, 1 died at 6 months untested |
| Family B | Parents: 1 deceased, 1 HIV-positive<br>No siblings | Parents: 1 untested, 1 HIV-positive<br>Siblings: 1 untested | Spouse HIV-positive<br>Children: 1 HIV-negative |
| Family C | Parent: single HIV-positive mother<br>Siblings: 5 HIV-negative | Parents: 1 deceased, 1 HIV-positive<br>Siblings: 3 untested, 1 HIV-negative | Single HIV-positive mother<br>Children: 1 HIV-positive |
| Family D | Parents: 1 deceased, 1 HIV-positive<br>Siblings: 2 untested | Parent: single HIV-positive mother<br>Siblings: 1 HIV-negative | Spouse untested<br>Children: 2 HIV-positive, 1 HIV-negative |
| Family E | Parents: 1 deceased, 1 HIV-positive<br>Siblings: 1 HIV-negative, 1 deceased | Parents: 1 deceased, 1 HIV-positive<br>Siblings: 2 untested | Spouse HIV-positive<br>Children: 2 HIV-positive, 1 HIV-negative |
| Family F | Parents: 1 deceased, 1 HIV-positive<br>Siblings: 1 untested | | Spouse deceased and untested<br>Children: 1 HIV-positive, 1 HIV-negative |
| Family G | Parent: single HIV-positive mother<br>No siblings | | Spouse HIV-negative<br>Children: 1 HIV-positive, 1 untested |
| Family H | | | Spouse deceased and untested<br>Children: 1 HIV-negative, 1 HIV-positive, 2 untested |
| Family I | | | Spouse HIV-positive<br>Children: 2 HIV-positive |
| Family J | | | Spouse HIV-positive<br>Children: 1 HIV-positive, 1 HIV-negative |
| Family K | | | Spouse HIV-positive<br>Children: 2 untested |
| Family L (married couple) | | | Married HIV-positive couple<br>Children: 2 HIV-negative, 3 untested |
| Family M | | | Spouse HIV-positive<br>Children: 1 HIV-negative, 1 untested |
| Family N | | | Spouse HIV-positive<br>Children: 2 HIV-positive, 3 HIV-negative, 1 stillbirth |
| Family O | | | Spouse deceased and untested<br>Children: 1 HIV-positive, 1 untested |

full disclosure of these HIV statuses. Some participants spoke of the need to live positively and move forward with their lives despite being HIV-positive or living with a parent with HIV. Frank, an HIV-positive boy stated:

> *At the beginning it was* [pauses, repeats the same words] *I couldn't take it, I couldn't believe it, and I was very depressed but as time went on I realized there was nothing I could do, so I had to take it, and I started behaving positively…After sometime I saw that I am*

*not different from the ordinary person. So I took it and I started taking my medication and now from that time I have learnt to live positively and that's why I am here.*

Many HIV-positive parents, HIV-positive children, and HIV-negative children expressed appreciation of the availability of medications (ART, cotrimoxazole, and multivitamins) for treatment, because the medications allowed them and their parents to remain healthy. HIV-positive parents and HIV-positive children who had other HIV-positive family members explained that they took their medications together, a practice encouraged by healthcare professionals to improve adherence. Brian, a healthcare professional explained:

*You find that even kids remind the parents, and it is like a communal thing, a family affair about ensuring that they all take their drugs, unlike when their parents are hiding, and they are fearing the child to know, so it kind of becomes a support thing in the family.*

Many participants expressed that an HIV status was perceived as being spread through a promiscuous lifestyle and that HIV-positive persons and their families were poorly regarded by community members. HIV-positive parents were especially ashamed about their HIV statuses because it had come about due to unprotected sex. HIV-positive and HIV-negative children expressed high awareness of the disease and how it was spread; some who were teenagers explained that this had caused them to delay initiating sexual activity. HIV-positive parents, HIV-positive children, and HIV-negative children spoke of the need to hide an HIV status because of high levels of gossip, misconceptions, stigma, and discrimination held by community members. HIV-positive parents were especially hesitant to disclose to their children because they might tell others and expose the family to stigma and discrimination. Healthcare professionals explained that these factors added to the challenges of HIV disclosure, living with an HIV status, or having an HIV-positive parent. Janet, an HIV-positive parent explained:

*The problems come in the neighborhood, this people you know they are strangers and they don't know you but once they find out what is happening to you, then you don't know what will happen. You can even find someone pouring water at your door, so you don't mention because somebody will know something to pin you down.*

### Evolution of disclosure

During clinic visits, healthcare professionals counseled HIV-positive parents that their children would eventually need to receive full disclosure of their own and/or their parents' HIV statuses. Stella, a healthcare professional stated:

*Usually what happens in disclosure, most of the time we always make the parents aware of this, that a child will definitely want to know how they got the infection. So we always advice the parents it is good for you to know your status so that should he or she ask then we can think of what to say. But if you don't know your status then it becomes very hard for us to explain how he got the infection and it will be very hard for the child to accept the status when he does not know where it came from.*

However, HIV-positive parents felt the need to prepare at their own pace and fully disclose to their children when they were ready for disclosure. HIV-positive parents diagnosed within 0–3 years of study participation were still secretive about their HIV statuses, and many had not yet thought about fully disclosing to their children. HIV-positive parents at 4–5 years post-diagnosis had started exploring and began making disclosure plans to be implemented at some time in the future; most had told some or all their children limited details about HIV.

HIV-positive parents who were beyond 5 years since diagnosis reported being at advanced phases of the disclosure process. Many had fully disclosed to their oldest children and also told their younger children limited details about HIV. These parents were ready to fully disclose to these younger children when they judged them to be ready. Paul, an HIV-positive parent explained the disclosure process to his children as follows:

> *When she* [oldest daughter was about eight-years-old], she *was first cheated that during her mother's illness, a blood transfusion was given at the hospital and that is where she* [mother] *got the problem. Afterwards, my wife and I had to sit down both of us to talk, look for the consequences, how these children were going to welcome this and the community. You know when children are told something new, some discuss it, they tell their friends they have this, so we had to sit down tell them* [limited details about HIV] *…When she* [oldest daughter] *was around 10, her mother took her to a clinic with her brother and sister. They were tested, the youngest two were confirmed positive, she was negative. They had a counselor who told them everything because her mother asked him is it possible for us to tell these children? The counselor said tell them.*

At the time of recruitment, we sought HIV-positive parents with children ranging in age from 8 to 17 years of age who had no, partial, and full disclosure of a family member's HIV status. The HIV-positive parents recruited into the study had a total of 37 living children varying in age from 4 to 26 years. In total, there were 49 children (including the seven HIV-positive and five HIV-negative children who participated) from all families represented in the study. Among these 49 children, those diagnosed HIV-positive in their teenage years received full disclosure of both their own and their parents' HIV statuses soon after HIV testing in a clinic setting from their parents or an older sibling with the help of healthcare professionals. Hannah, a 16-year-old HIV positive girl explained her recent diagnosis the following way:

> *I just started feeling cold, then the following morning I convulsed, and from my waist downwards it's like* [clicks tongue] *I just couldn't walk, I don't know what happened. So I was brought here to Kenyatta and I was admitted. There was a counselor who used to come* [clicks tongue] *and started asking me questions. Now those questions were like uh, if you can be told what you are suffering from how could you feel. I told him me I know I am suffering from malaria and convulsing. So me I just started feeling this guy I dont know what he wants from me, so I started hating him. Then one day curiosity told me to steal my file, and I read that my mother passed on with AIDS, gosh! Then I just kept quiet, I started asking myself then if my mother died of AIDS then how come I am negative? And then I*

*started remembering when my sister died, people were saying she died of pneumonia, then how comes she died of pneumonia?* [Clicks tongue] *Uh I just ignored and I said maybe this is not true. After I was discharged, I used to see my sister cry sometimes, my cousin started showing me some love that I have never been shown* [clicks tongue]. *Uh, then I started saying uh, maybe it's just because I was from the hospital. Then one day my sister told me she is taking me somewhere, she took me to a support centre. There is a counselor there, she asked me what I know my mother suffered from? I told her, then my sister got shocked coz she thought I never knew and then* [clicks tongue], *the counselor got worried. She started telling me how people suffer what blah, blah, blah. I just told her to stop giving me lectures and tell me the truth. She told me I am HIV-positive.*

Children diagnosed HIV-positive before adolescence were enrolled into care and received counseling and partial disclosure over time. When judged ready, they received full disclosure of either their own and/or their parents' HIV statuses at home or in the clinic from their parents, relatives, and/or healthcare professionals. HIV-negative children in the study sample, and the HIV-negative and untested children of the HIV-positive parents in the study sample also received increasing details of their parents' HIV statuses over time until they were fully disclosed to at home by their parents or relatives. Ben, an HIV-negative child recounted his disclosure process as follows:

*Let me start back from about 2002* [when he was about six-years-old] *there to 2005, my mum got sick of meningitis after my brother was born, then she recovered but not completely… So this year* [when he was 14-years-old], *we were just telling ourselves some back stories. Then I started talking to her* [mother] *that you know I took care of my brother when you were in the hospital. Then we went on talking and she told me that you know that I* [mother] *am positive, I have HIV. I was not that surprised coz even there was a time that I looked in her files* [when about eight-years-old] *and then I saw something about HIV-positive or something. I did not even concentrate on it; I was looking for my birth certificate that she had put in her files. So when she told me, I remembered that I saw a certain paper, positive HIV.*

During their interviews, all participants were asked the ideal age for disclosure to children; they generally thought that disclosure should not occur for children between 0 and 4 years of age; those between 5 and 9 years should receive partial disclosure; and they thought that full disclosure should occur for those 10 years and above. HIV-positive parents prepared and disclosed to their children differently, without an identifiable pattern, or regard to HIV status, but mainly based on birth order. In a few families, children close in age received disclosure at the same time. This resulted in children of diverse ages being at different disclosure levels within the represented families; there was a child as young as four years with full disclosure and a 17-year-old child who did not live at home with no disclosure. Doris, an HIV-positive parent, chose to have her sister disclose to her two sons at the same time. She explained:

*I was not courageous enough to tell them* [11-year-old HIV-positive and 9-year-old HIV-negative sons]. *My sister told them so that they can be supporting me whenever I send*

*them they go. I thought she was the best person because I am close to her. I thought it was good for them to know about my oldest son's and my HIV statuses because by that time my son was not taking ARVs. So she had to tell the boy so that he can be keen on taking the drugs, he was not taking his drugs very well.*

## Questions associated with disclosure

The HIV disclosure process was accompanied by many questions throughout. During disclosure preparation, healthcare professionals counseled parents to anticipate and address children's questions before, during, and after partial or full disclosure. HIV-positive and HIV-negative children with partial disclosure explained that they had many questions about theirs' or their parents' HIV statuses, why they or their parents were taking medications and/or attending clinic visits, and what had caused the death of family members. HIV-positive parents explained that their children with partial disclosure had similar questions as the child participants in the study, and that these questions made parents consider whether it was the right time to start preparing for, or implementing, the full disclosure plans they had in place. Some healthcare professionals stated that they usually counseled parents that when children were asking pointed questions such as these, the children were ready for full disclosure. However, if a child was deemed as not ready for full disclosure, age-appropriate answers were provided. Two healthcare professionals stated:

*I think the time will come when the child starts to ask questions. I don't think you should give the information until the child asks questions. So when the child asks the questions then you give the age-appropriate answers. So I will still use the language that the child asks me, the language that they would understand.* [physician]

*Sometimes the parents actually feel or still to them, it doesn't seem that the child actually wants to know. The child is just being inquisitive or the child is being naughty, but yet we tell them, actually if they are asking about the illness, it is the right time for them to know.* [psychologist]

At the time of full disclosure, a few HIV-positive parents expressed that they had asked their children what they knew about HIV and their children's views about HIV-positive persons. The answers given by their children helped HIV-positive parents phrase the disclosure news. HIV-positive and HIV-negative children expressed that at the time of full disclosure, they had wanted to know about the origin of the illness, who else was infected in the family, and what had caused the deaths of family members. HIV-positive parents expressed that their children with full disclosure also had similar questions. Caroline, an HIV-positive parent stated the following about her son's questions during the disclosure process:

*He was really asking a lot of questions like what killed my father? Because that was in him, he used to ask so many questions about how did he die, where he was sick, what was it exactly? If you say the chest, he would ask even me when I get chest pains I will also die?... When I was planning to tell him, I wanted him to know about his sickness so that*

*he becomes serious about his medication, because he used to ask questions why am I taking medicine and I am not sick?* [Following full disclosure] *he asked who else knows and I told him my cousin, one of my sisters and my brothers knows, and also his grandparents on both sides. So he said why did you tell them even before you told me? So I told him there is a time we were sick, we had to buy medication, and they had to support us, and he understood.*

Following initial diagnosis and/or full disclosure, HIV-positive parents and HIV-positive children expressed that they infrequently asked themselves where, why, and how they had become infected; HIV-negative children wondered how their parents became infected. Some healthcare professionals and HIV-positive parents expressed that following full disclosure, most questions from children ended. Those that remained included how long children and/or their parents would continue to take medications, if there was a cure for HIV, and clarification/confirmation about the origin of the illness. HIV-positive parents expressed that questions about the origin of the illness made them very uncomfortable although they answered them the best they could. Healthcare professionals expressed that these questions from children made HIV-positive parents regretful of having fully disclosed. HIV-positive and HIV-negative children with full disclosure expressed that they asked these questions in order to obtain answers for issues that were still in their minds or bothering them. Margaret, an HIV-positive parent explained:

*Somehow, it is difficult, it is 50/50, either you are relieved, sometimes it also disturbs you* [laughs]. *Now maybe you tell your child but you don't know if she has received it well. You wonder why did I do it because maybe of some funny behaviors. Maybe she asks you a lot of questions unnecessary, you feel uh, this one why maybe did I tell her, maybe that's what has brought all those questions, but it is good to tell her...She* [HIV-positive daughter with full disclosure] *can sit down and ask you, mum me now I am positive, you also you are positive and you said after daddy died you were just this way, how did you get this thing? You just tell her maybe I don't know. Now what about me, what can I answer if somebody asks me how did I get this thing? I get it from my mother or what? So at those questions you have nothing completely...I was just answering some of them* [laughs], *ignore some because at times some were too much, too difficult to answer.*

## Emotions associated with disclosure

The HIV disclosure process was accompanied by varied positive and negative emotions throughout. Before full disclosure, HIV-positive parents felt guilt and shame about their role in bringing HIV into the family. They worried about their capability to fully disclose, and many feared how their children would react when fully disclosed to, or that they would in turn tell other people. Before full disclosure, HIV-positive and HIV-negative children, as well as children of HIV-positive parents, sensed secrets around them and worried, speculated, and were suspicious about theirs and/or their parents health statuses, prior deaths in the family, and the reasons for medication consumption, clinic attendance, and blood draws. Jack, an HIV-positive boy, explained his pre-disclosure suspicions this way:

*They* [mother and healthcare professionals] *didn't tell me* [the reasons for HIV testing], *they told me it is for malaria, but I wondered why not this finger? Why a lot of blood? But they did not tell me. Even when she was being told the results they told me first of all to go out. Then I started the medicine, by that time they told me to take the medicine for coughs, but one week later I refused to take them. I asked, I insisted to be told they are for what because they were a lot of medicines.*

At the time of partial or full disclosure, HIV-positive parents explained that they mostly experienced relief from finally explaining the illness. A few who fully disclosed felt guilty and confused when their children received the news poorly. Some expressed that they had reassured their children and provided them with hope that things would be okay. HIV-positive and HIV-negative children expressed that they had also felt relief about receiving an explanation of what was wrong with them and/or their parents. However, they also experienced a range of negative emotions including fear, anger, disbelief, shock, distress, confusion, sadness, unhappiness, hopelessness, depression, and crying. HIV-positive parents expressed that their children had similar emotions at the time of partial or full disclosure; one HIV-positive parent expressed that her teenage daughter was suicidal. Healthcare professionals confirmed that these were typical emotions expected at the time of partial or full disclosure. Samuel, an HIV-positive child stated:

*First of all, you know the way the doctor told me that I am HIV-positive, I felt like my heart was beating, beating very fast because I knew that I was going to die... I just felt sad, angry, confused, and hopeless, so I wanted the doctor to tell me where and how my life would go. So he explained to me that this is not the end because it is just the beginning, so don't be so scared about this disease.*

In the weeks after partial and full disclosure, HIV-positive children, HIV-negative children, and HIV-positive parents along with their children, experienced alternating positive and negative emotions. HIV-positive parents expressed that they experienced relief followed by unhappiness, depression, guilt, regret, and uncertainty when they observed their children acting differently because of disclosure. HIV-positive and HIV-negative children continued to feel relief, but they also cried and were depressed, hopeless, sad, unhappy, and a few HIV-positive children experienced self-hate. Sarah, an HIV-positive parent, expressed the following about the weeks after disclosure:

*I was relieved to have told him* [teenage HIV-positive son] *but they start asking questions, like him he asked many questions. When we came here, he will start staring at people, he sees another one with maybe a big chest, such like things, mmm, very sunken cheeks, such like things. And then he will want to know, if I keep on taking these drugs will I be like so and so?... I needed counseling especially when I saw him, umm being different, now he was just quiet, I didn't know what he was thinking and what he wanted to do. I needed to know how to make him at least open up to me what he was thinking. It was very important, I didn't want things to turn out bad, some turn out to be suicidal, I didn't want such.*

At the time of their interviews, HIV-positive parents, HIV-positive children, and HIV-negative children were asked how they were currently feeling. A few expressed hope

that a cure would be found for HIV and many stated their families were closer as a result of disclosure. Some expressed that when they had family problems or health issues going on in themselves or a family member, they re-experienced the negative emotions they had felt at the time of partial or full disclosure. Ben, an HIV-negative child stated:

*Sometimes when I recall that she* [mother] *is HIV-positive, I just feel sad, upset, hopeless, sometimes but not for a long time. If I see that she is unwell, because sometimes she tells me that she is not feeling well, I just start to wonder what is wrong? How can I help? Then I just don't know what to do.*

## Benefits and consequences of disclosure

All participants provided many parent and child-related benefits of disclosure. The benefits for HIV-positive parents included less stress from hiding the illness, improved psychological health, increased support from their children, increased ability to take medications and attend clinic visits openly, improved medication adherence, and increased bonding with their children. For HIV-positive children, the benefits included increased independence; improved self-care, self-medication, and medication adherence; and a greater understanding about their HIV statuses, medications, and clinic attendance. For HIV-negative children, the benefits included increased bonding with their parents, awareness of the disease, acceptance of HIV-positive persons, access to HIV testing, and fewer secrets within the home. In addition to other benefits aforementioned, healthcare professionals especially advocated for increased disclosure levels so that stigma and discrimination among community members would be reduced. Jack, a healthcare professional explained the benefits of disclosure this way:

*When minors are disclosed to, they give that full support. The children will tell you mum you have not taken ARVs, mum you want water, when you are down then they feel for mum's needs and that psychological problem now does not come fully because they are a bit prepared…Disclosure promotes adherence to ARVs, because where a child does not know then they will also ask why am I taking medicine and yet I am not sick? As children grow more mature and they interact in school with other peer friends and colleagues, they should know when they are HIV-positive so that if they get into immoral behavior, then they are at risk of infecting others.*

Some participants also provided accounts of negative effects that had resulted from disclosure. Some HIV-positive parents expressed that they had experienced increased stress when their children reacted negatively to disclosure, were angry or blamed them, and/or were disrespectful towards them. Emma, an HIV-positive woman separated from her husband. She stated:

*When I knew my status, I was married and I was pregnant, that's what made me to know my status. I went and told my husband, he told me that's mine, he doesn't know about it. So I stayed with my husband for about one year, he used to talk bad to me, we* [her sons and her] *didn't have any peace in the house but before that he was okay. So I had to decide to go, and that's when we divorced. I went back to my parents and I stay with them until now.*

Some of the negative effects that occurred in children following disclosure included drops in school performance among HIV-positive children; for HIV-negative children, there was a loss of normalcy in their everyday lives, and adoption of more responsibility around their homes. Older children who received full disclosure experienced increased stress from keeping the illness a secret from their younger siblings.

## HIV disclosure model

We propose a new HIV disclosure model displayed in Fig. 2 for use in disclosing a parent's and/or a child's HIV status within HIV-affected families. The model takes into account the six themes that emerged from our data; it shows that HIV testing, living with HIV, evolution of disclosure, questions associated with disclosure, emotions associated with disclosure, and the benefits and consequences of disclosure continuously interplay with each other throughout the HIV disclosure process. The model also demonstrates the challenging and cyclical nature of the HIV disclosure process. Like *Lowenthal's et al.'s (2014)* pediatric HIV disclosure model, the proposed model is a clinic-based healthcare professional driven model in which clinic staff provide regular disclosure-related assessments and counseling/advice to HIV-positive parents and HIV-positive children during the routine clinic visits they attend together. After each assessment, our model calls for healthcare professionals to determine what phase HIV-positive parents and HIV-positive children are in, then provide targeted advice to assist them to progress to the next phase of the disclosure process until full disclosure of a parent's and a child's HIV status occurs. Healthcare professionals also assist HIV-positive parents in determining whether full disclosure should occur at the clinic for parents who need assistance, or at home for those capable of fully disclosing on their own. Finally, to assure the best possible outcome, the model calls for post-disclosure follow-up assessments of both the parent and child.

## DISCUSSION

The majority of the 33 million people infected with HIV are adults of childbearing ages (*UNAIDS, 2014*); perinatally infected children are beginning to have children of their own (*Corbin, 2008*; *Mitchell et al., 2008*). Therefore, the urgent need for guidelines and models for HIV disclosure of a parent's and/or a child's HIV status will persist for years to come. In this study, we described the entire HIV disclosure process from the point of diagnosis of an index person within Kenyan HIV-affected families, through disclosure of a parent's and/or a child's HIV status to one or more children in the household. We also present a new model of HIV disclosure that incorporates our six themes that span the HIV disclosure process; these include HIV testing, living with HIV, evolution of disclosure, questions associated with disclosure, emotions associated with disclosure, and the benefits and consequences of disclosure.

The consequence theory of HIV disclosure and the disclosure process models previously proposed in the literature are geared for disclosure of a parent's HIV status (*Chaudoir, Fisher & Simoni, 2011*; *Serovich, 2001*; *Qiao, Li & Stanton, 2013*), while the four-phase model (*Tasker, 1992*), the *Blasini et al. (2004)*, and the *Lowenthal et al. (2014)* models are geared for disclosure of a child's HIV status. Many HIV-affected families have both parents

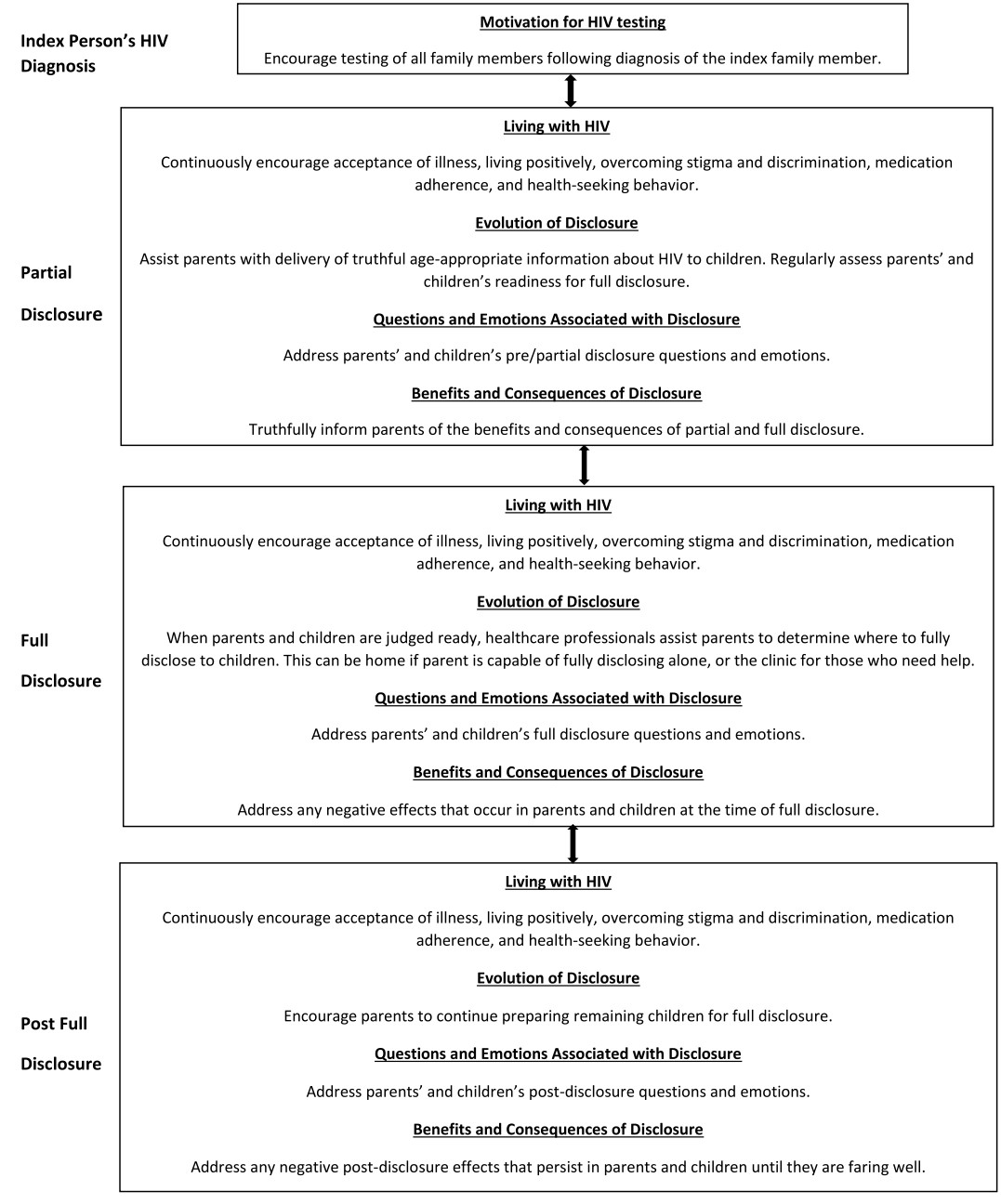

**Figure 2  HIV disclosure model.**

and children who are HIV-positive (*Gachanja, Burkholder & Ferraro, 2014a*; *Gachanja, Burkholder & Ferraro, 2014b*; *Republic of Kenya, 2009*); our proposed model is well suited for disclosure of both parent's and child's HIV statuses within HIV-affected families. Unlike the other aforementioned models that prepare children for disclosure of their own HIV statuses (*Blasini et al., 2004*; *Lowenthal et al., 2014*; *Tasker, 1992*) or disclosure of their parents' HIV statuses (*Chaudoir, Fisher & Simoni, 2011*; *Qiao, Li & Stanton, 2013*; *Serovich, 2001*), our model provides for simultaneous preparation of parents and children for

disclosure of both their illnesses by healthcare providers; such disclosure happens through the provision of targeted advice/services that sequentially move parents and children together through the pre, during, and post-disclosure phases. Parent-child(ren) dyad barriers, consequences, and benefits of disclosure are evaluated regulary (*Serovich, 2001*) to maximize optimal outcomes (*Chaudoir, Fisher & Simoni, 2011*; *Qiao, Li & Stanton, 2013*). The model also calls for healthcare professionals to help HIV-positive parents determine the most suitable setting for full disclosure to their HIV-positive children; however for best outcomes, *Kidia et al. (2014)* have advocated for teenage children to receive disclosure in a clinic setting. The *Blasini et al. (2004)* model calls for post-disclosure follow-up of the child; our model posits the importance of follow-up with both parents and children until is assured that they are faring well.

HIV testing of all family members, disclosure, and linkage to care are important because they assure that all infected persons in a family are enrolled into care early enough to achieve positive treatment outcomes (*Luchters et al., 2008*; *Rosen & Fox, 2011*). Even when they test negative or remain untested, for the wellbeing of all HIV-affected family members, it is important to include HIV-negative and untested children in HIV disclosure activities. HIV-negative children have previously expressed a desire to be included in disclosure-related activities (*Gachanja, 2015*). Policymakers and HIV clinics should put in place services/programs that include HIV-negative and untested children, either during routine clinic visits attended by their HIV-positive parents and HIV-positive siblings, through special appointments with trained HIV disclosure counselors, or within support group settings. Additionally, provision of routine stigma reduction counseling during clinic visits helps HIV-positive parents and their children positively deal with and handle the high stigma/discrimination levels present in high prevalence countries (*NASCOP, 2014*). The model posits that such counseling would increase disclosure rates and improve the outlook for HIV-affected family members. Our model of HIV disclosure should guide HIV disclosure activities soon after the diagnosis of an index family member within HIV-affected families. Using the model as a guide, healthcare professionals create a family-oriented plan geared at HIV testing and counseling of all family members, followed by HIV disclosure to all children within the family in a targeted, timely, truthful, and culturally age-appropriate manner.

This study's results reinforce findings from other studies completed in Kenya that have shown that HIV-positive parents are challenged with disclosure to their partners/spouses (*Rogers et al., 2015*; *Walcott et al., 2013*) and their children (*Gachanja, Burkholder & Ferraro, 2014a*; *Gachanja, Burkholder & Ferraro, 2014b*; *John-Stewart et al., 2013*; *Turissini et al., 2013*; *Vreeman et al., 2015*; *Vreeman et al., 2010*). HIV testing of their children is also a challenge (*Rwemisisi et al., 2008*; *Ishikawa et al., 2010*). Our HIV-positive parents reported that it took years to test their spouses and children and to fully disclose to their children; some diagnosed within 3 years of study participation were still secretive about their HIV statuses. Prior researchers have reported that children are emotional and ask questions in the immediate period following full disclosure of their own (*Vaz et al., 2010*) and their parents' HIV statuses (*Kennedy et al., 2010*). Our findings show that children's questions persist long after full disclosure and make parents very uncomfortable. Additionally,

HIV-affected family members' emotions appear to swing between positive and negative states, or recur when family issues or sickness are encountered. Because HIV disclosure is a continuous process, routine use of our model during each clinic visit will determine what difficulties an HIV-affected family is encountering; healthcare professionals will then be able to offer any needed assistance to counteract these difficulties.

Adults and children in Kenya continue to practice unsafe sexual behaviors despite high awareness of HIV (NASCOP, 2014). However, none of the teenage children (especially HIV-positive ones) in this study were sexually active, possibly indicating that HIV awareness programs may be resonating with some children. Sex education should be incorporated into routine clinic visits for HIV-affected family members; this would help curb the incidence of new infections among HIV-negative and untested children in these households, and prevent the spread of HIV from HIV-positive children to their peers. As noted in our findings, two children received inadvertent full disclosure through medical records. Therefore, healthcare professionals should take measures to ensure clinic files are stored securely to prevent inadvertent disclosure to children; HIV-positive parents should also be cautioned to keep HIV-related documents securely stored at home.

Our study is not without limitations. We did not interview parent–child dyads; therefore, the results may not mirror the experiences of HIV-positive parents and children within similar HIV-affected families. Our overall sample size of 34 participants consisted of smaller subgroups of HIV-positive parents, HIV-positive children, HIV-negative children, and healthcare professionals, whose perspectives may not represent those of the populations they signify. However, the study results provide a good overview of the pre, during, and post phases of the HIV disclosure process from the perspectives of HIV-affected family members and the healthcare professionals who provide them with disclosure-related services. We recommend that future researchers test the viability and feasibility of our HIV disclosure model in resource-poor and rich nations. The model's utility should also be tested for use in prepration of non-parental caregivers (e.g., grandparents, aunts, uncles, siblings, etc.) for HIV disclosure of HIV-positive children's statuses.

## CONCLUSION

HIV disclosure of a parent's and a child's HIV status is challenging for healthcare professionals; and also HIV-positive parents and their HIV-positive, negative, and untested children. At this time, Kenya does not have guidelines or models for HIV disclosure of a parent's or a child's illness; prior researchers have called for guidelines creation to ease facilitation of HIV disclosure (Gachanja, Burkholder & Ferraro, 2014a; Gachanja, Burkholder & Ferraro, 2014b; John-Stewart et al., 2013; Turissini et al., 2013). We present an HIV disclosure model to be used by healthcare professionals in Kenya and other similar cultures, for disclosure of a parent's and a child's HIV status to HIV-positive, negative, and untested children within HIV-affected families.

## ACKNOWLEDGEMENTS

We thank the participants for sharing their disclosure experiences with us. We would like to thank Dr. Charles Kabetu, Dr. James Kiarie, Ms. Mugambi, Ms. Gacheru, Nelly Opiyo, Godfrey Mureithi, David Mutabari and KNH CCC staff for their assistance during the conduct of this study. We would also like to thank Purity Wambui Kibino for her assistance with data transcription and Opeyemi Fasina for copy editing of the manuscript. Finally, we would like to greatly thank Dr. Aimee Ferraro for her advice and mentoring during the conduct of this research study and preparation of the manuscript.

### Funding

The authors received no funding for this work.

### Competing Interests

Gary J. Burkholder is an employee of Laureate Inc.

### Author Contributions

- Grace Gachanja conceived and designed the experiments, performed the experiments, analyzed the data, wrote the paper, prepared figures and/or tables, reviewed drafts of the paper.
- Gary J. Burkholder analyzed the data, wrote the paper, reviewed drafts of the paper.

### Human Ethics

The following information was supplied relating to ethical approvals (i.e., approving body and any reference numbers):

Ethics approval was received from the Kenyatta National Hospital Research Standards and Ethics Committee (Approval # P373/10/2010) in Nairobi Kenya, and the Walden University Institutional Review Board (Approval # 11-10-10-03904).

### Data Availability

Participant codes have been provided in the manuscript.

### Supplemental Information

Supplemental information for this article can be found online at http://dx.doi.org/10.7717/peerj.1662#supplemental-information.

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
