# Peer review of "A model for HIV disclosure of a parent’s and/or a child’s illness"

_PeerJ, doi:10.7717/peerj.1662_

## Round 0.1 · original submission · Minor Revisions

I have obtained now three reviews of your paper and the reviewers feel it has improved, but still requires a bit more work before acceptance. All three provide a number of comments that will be important to address, especially the 'major' issue listed by reviewer 3 with respect to the novelty of the study.

Reviewer 1 ·

Basic reporting

No Comments

Experimental design

No Comments

Validity of the findings

No Comments

Additional comments

This paper, presenting results from a qualitative study, presents a model for HIV status disclosure and suggestions for healthcare workers to effectively assist in this process. Overall, the paper is well-written. The qualitative methodology is appropriate and well reported. There are in-text comments in the manuscripts and a few additional points below that I believe would improve the paper:

• Defining the distinction between disclosure to children and onward disclosure needs to be more distinct. If the authors claim that these processes are very intertwined (which they are), then they need to still acknowledge this difference.
• introduction section needs to do a better job of referencing the newer, evolving literature on disclosure. There have been several good examples of excellent qualitative and quantitative research over the last 2 years that look at the question of disclosure in a similar samples in other SSA countries and have been omitted by the authors. Please rerun the search and include this literature new and relevant literature including:
o Vreeman, Rachel C., et al. "Disclosure of HIV status to children in resource-limited settings: a systematic review." Journal of the International AIDS Society16.1 (2013).
o Evangeli, Michael, and Ashraf Kagee. "A model of caregiver paediatric HIV disclosure decision-making." Psychology, health & medicine (2015): 1-15.
o Lowenthal, Elizabeth D., et al. "Disclosure of HIV status to HIV-infected children in a large African treatment center: Lessons learned in Botswana." Children and Youth Services Review 45 (2014): 143-149.
o Tadesse, Birkneh Tilahun, Byron Alexander Foster, and Yifru Berhan. "Cross Sectional Characterization of Factors Associated with Pediatric HIV Status Disclosure in Southern Ethiopia." PloS one 10.7 (2015): e0132691.
o Kidia, Khameer K., et al. "HIV status disclosure to perinatally-infected adolescents in Zimbabwe: a qualitative study of adolescent and healthcare worker perspectives." PloS one 9.1 (2014).
o Papers contained in AIDS Care Special Issue on Disclosure (2015): http://www.tandfonline.com/doi/full/10.1080/09540121.2015.1102687
o Papers contained in AIDS Special Issue on Disclosure (2015): http://journals.lww.com/aidsonline/toc/2015/06001

• The results section needs to make a trade off between claims and evidence. It is always helpful to have illustrating quotations – consider putting these in a table if there are too many, or else include only data that is most relevant to the research question and which emerged most strongly from the analysis.
• The model itself is a somewhat weak – a mixture of results and ideas, that could probably be exchanged for a well written “policy box”, similar to the suggestions already existing. The authors make no effort to reference previous models and describe how theirs compares (see abovementioned, and further literature).

Annotated reviews are not available for download in order to protect the identity of reviewers who chose to remain anonymous.

Reviewer 2 ·

Basic reporting

No comments

Experimental design

No Comments

Validity of the findings

- in the methods section it is not clear how the indexing of clients came about; how were those clients selected as the index clients. a bit of an explanation will be important
- Also in the methods section the order of different action in the selection of participants need to be consistent i.e what begun first to the end. this will make it easy to follow
- in the results section where the element of disclosure is associated with level of awareness about HIV and in turn delayed sexual debut among teenage children, is an important finding that may require further illumination in the discussion (this is found between line 179-181)

Additional comments

- the reference of HIV as an illness is misleading. The manuscript will read better if 'illness*' is replaced with 'HIV status*'
- results too long at some repetition is observed. consider merging 'emotions related with disclosure' theme with 'benefits and consequences of disclosure' then summarize and bring out key points worth taking to the discussion.
- discussion seem not to follow the inverted funnel kind of approach where main results are highlighted followed with relation to existing literature and implication to policy and practice.

·

Basic reporting

The submitted article is a qualitative study of a small number of HIV-positive parents, HIV-positive children, HIV-negative children, and healthcare professionals in Kenya, designed to understand the process of disclosure. The manuscript is fairly well written and easy to follow. However, there are a few minor issues that need to be addressed, and two more major issues. These will each be briefly discussed.

1. The number of participants approached to participate in the study and the refusal rate are not provided. Given how small the sample size was, it is critical that this information be provided, as it provides some information on potential generalizability of the findings.

2. One major issue is that the paper doesn’t really provide anything new to the literature. At this point in time, there is a fairly rich literature surrounding disclosure of HIV status, especially parental disclosure to children. These findings have pretty much been reported previously in low resource countries, as well as the U.S., the U.K., etc. The authors need to point out early in the Abstract and Introduction what this study adds to the literature. Perhaps some content in the Discussion on the similarities and differences of the findings in Kenya relative to all of the literature on this topic area that precedes this would be useful.

Experimental design

3. It would be more helpful if a few more quotes were added into the Results section, as few are given for each topic area.

Validity of the findings

4. The most problematic issue is that following the Introduction, the authors simply announce that based on the results of their study they are proposing a new HIV disclosure model. There is absolutely no discussion of current models, or what constructs from already available literature might be guiding them in the development of a new model other than their own small sample findings. There have been several disclosure models available for years, in fact, there has been a randomized controlled trial of a disclosure intervention for HIV-positive mothers living with HIV/AIDS that had one of the published disclosure models as a framework that was adapted for families. The authors need to attempt some background of models in the Introduction. Just as importantly, they need to discuss whatever “new” model they are proposing in the context of what it brings to the literature relative to the several other disclosure models that are already published. This could be in interesting area of what constructs are fairly stable across models, and why in Kenya something different might be needed.

---

## Round 0.2 · accepted · Accept

Thank you for your careful consideration of the constructive reviews. I agree that your paper is now significantly improved and appropriately addresses the reviewers' concerns. I think it is a solid contribution!